# Marine Bioactive Compounds against *Aspergillus fumigatus*: Challenges and Future Prospects

**DOI:** 10.3390/antibiotics9110813

**Published:** 2020-11-16

**Authors:** Chukwuemeka Samson Ahamefule, Blessing C. Ezeuduji, James C. Ogbonna, Anene N. Moneke, Anthony C. Ike, Bin Wang, Cheng Jin, Wenxia Fang

**Affiliations:** 1National Engineering Research Center for Non-Food Biorefinery, Guangxi Academy of Sciences, Nanning 530007, Guangxi, China; chukwuemeka.ahamefule.pg79618@unn.edu.ng (C.S.A.); bwang@gxas.cn (B.W.); 2College of Life Science and Technology, Guangxi University, Nanning 530005, Guangxi, China; 3Department of Microbiology, University of Nigeria, Nsukka 410001, Enugu State, Nigeria; james.ogbonna@unn.edu.ng (J.C.O.); anene.moneke@unn.edu.ng (A.N.M.); anthonyc.ike@unn.edu.ng (A.C.I.); 4Department of Microbiology, University of Jos, Jos 930001, Nigeria; bezeuduji@gmail.com; 5State Key Laboratory of Non-food Biomass and Enzyme Technology, Guangxi Academy of Sciences, Nanning 530007, Guangxi, China

**Keywords:** marine resources, *Aspergillus fumigatus*, bioactive compounds, screening model, fungi

## Abstract

With the mortality rate of invasive aspergillosis caused by *Aspergillus fumigatus* reaching almost 100% among some groups of patients, and with the rapidly increasing resistance of *A. fumigatus* to available antifungal drugs, new antifungal agents have never been more desirable than now. Numerous bioactive compounds were isolated and characterized from marine resources. However, only a few exhibited a potent activity against *A. fumigatus* when compared to the multitude that did against some other pathogens. Here, we review the marine bioactive compounds that display a bioactivity against *A. fumigatus.* The challenges hampering the discovery of antifungal agents from this rich habitat are also critically analyzed. Further, we propose strategies that could speed up an efficient discovery and broaden the dimensions of screening in order to obtain promising *in vivo* antifungal agents with new modes of action.

## 1. Introduction

*Aspergillus fumigatus* is a saprophytic mold commonly found in the environment, with spores that are very light and easily disseminated [1]. It is also a potentially dangerous opportunistic pathogen that is reported as being the number one causative mold for mycoses, especially in immunocompromised patients [2]. Invasive aspergillosis (IA) caused by *A. fumigatus* is a severe systemic infection with high mortality and morbidity rates. IA has an annual global incidence of over 200,000, and the mortality rate could reach almost 100% among certain categories of patients if not properly treated [2,3,4].

With the already existing challenge of having only a limited repertoire of antifungal drugs for the treatment of aspergilloses, the continuous rise of drug resistance in *A. fumigatus* strains [5,6,7] further aggravates this challenge. Reports of *A. fumigatus* developing a resistance, in both clinical [8,9] and environmental [10,11] isolates, to antifungal drugs such as amphotericin B [12], azole-class drugs [13,14] and echinocandins [5,15], as well as developing a multidrug resistance [16], are increasing annually.

In the UK for example, Public Health England reported over a five-fold increase in *A. fumigatus* isolates that showed resistance to itraconazole (Minimum Inhibitory Concentration (MIC) ≥ 2.0 μg/mL) from 2012 (1.4%) to 2016 (8.5%), whereas 4.7% and 6.9% increased resistance to voriconazole (MIC ≥ 2.0 μg/mL) and posaconazole (MIC ≥ 0.25 μg/mL), respectively [17]. A study from Brazil by Reichert–Lima [18] showed one of the highest resistances (27%) of *A. fumigatus* to amphotericin B (MIC ≥ 2.0 μg/mL).

Due to its peculiar environment and rich biodiversity, the marine resource is a treasure box for isolating novel bioactive compounds. Over the years, antibacterial [19,20,21], antifungal [22,23], antiviral [24,25,26], anthelminths [27,28], antiprotozoan [29,30,31], antitumor [32,33], anticancer [34,35,36], anti-inflammatory [37,38], antioxidant [39,40], antiaging [41,42] and antidiabetics [43,44,45] compounds have been isolated from different marine organisms.

Among the reported antifungals isolated from the marine environment, only a few have been found to be effective against the recalcitrant fungus, *A. fumigatus.* Considering the threat of this important pathogen, in this review we have compiled and discussed bioactive compounds, isolated from the marine environment over the last two decades, which displayed an activity against *A. fumigatus*. We have also discussed the challenges restricting the discovery and isolation of effective agents against *A. fumigatus* from the marine environment. We have further proffered enhanced methods and multiple strategies that may improve the future discovery of marine antifungal agents.

## 2. *A. fumigatus* and Aspergillosis

Yu et al. [46] defined aspergillosis as a clinical infection caused by the genus *Aspergillus,* which can lead to allergic, superficial, saprophytic or invasive diseases. Cases of aspergillosis have constantly been on the increase since it was first reported over a century ago by Bennett [47]. Although aspergillosis is popularly associated with immunocompromised patients, over the years it has rapidly evolved in epidemiology, with increasingly new groups of patients at risk, even among the supposedly non-immunocompromised individuals [1,48].

Cases of varying kinds of aspergillosis have been reported among non-immunocompromised patients for years now, such as craniocerebral aspergillosis of sino-nasal origin [49,50], chronic invasive sinus aspergillosis [51], orbital aspergillosis [52,53,54], fatal invasive aspergillosis [55], intracranial aspergillosis [56], central nervous system aspergillosis [57], pulmonary aspergillosis [58,59], and adrenal and hepatic aspergillosis [60].

*A. fumigatus* has been reported as the most frequent etiological mold for invasive aspergillosis over the past two decades [61,62]. Unfortunately, the rapid evolution of drug resistant strains of this fungus complicates aspergillosis cases and limits clinical treatment by available antifungal drugs [6,8,63]. This has driven global searches for promising and potent new agents against *A. fumigatus.* Notwithstanding the fact that no new drug class has been discovered, some promising effective compounds have, however, been isolated and characterized from the marine environment.

## 3. Bioactive Compounds from Marine Organisms against *A. fumigatus*

A number of effective bioactive compounds against *A. fumigatus* have been isolated from marine organisms over the last two decades. Some of these compounds have been characterized and demonstrated as giving an appreciable potency against *A. fumigatus.* Other compounds, on the other hand, have been reported to have a weaker activity. These active compounds were isolated from different marine organisms, including bacteria [64,65], most of which were actinomycetes [66,67,68], fungi [69,70], algae and seaweed [71,72,73,74], sponges [75,76,77,78], and sea cucumbers [79,80,81].

### 3.1. A. fumigatus Effective Compounds from Marine Bacteria

Compounds with antifungal activity against *A. fumigatus* have been isolated from marine bacteria such as *Bacillus* [65,82], and more frequently from the unicellular filamentous bacteria actinomycetes [83,84,85]. Certain compounds, such as the caniferolides [85], have been characterized and shown to have a good activity against *A. fumigatus* (Table 1; Figure 1). Some other compounds, such as 30-oxo-28-*N*-methylikarugamycin, isolated from *Streptomyces zhaozhouensis,* only showed a very weak activity against *A. fumigatus,* with MICs of ˃64 μg/mL [83]. On the other hand, extracts from some streptomycetes [86,87], and other bacteria like *Micrococcus* sp., *Flavobacterium* sp. and *Streptomyces* sp. [64], have also been demonstrated as possessing very effective activities against *A. fumigatus* (Table 2). More interestingly, some of these extracts also showed promising activities against *A. fumigatus* strains with multidrug resistance (MDR) (Table 2).

### 3.2. A. fumigatus Effective Compounds from Marine Sponges

Marine sponges, such as *Theonella swinhoei, Siliquariaspongia japonica* and *Microscleroderma herdmani,* are other categories of interesting marine organisms demonstrated to produce effective antifungal compounds against *A. fumigatus.* Compounds such as Aurantosides E, A, B and Microsclerodermin B are the best antifungal bioactive compounds against *A. fumigatus*, with MICs of 0.04, 0.16, 0.16 and 0.6 μg/mL, respectively [77,88] (Table 3; Figure 2). Moreover, other Microsclerodermins (A, J and K) and Swinhoeiamide A also exhibit a good antifungal activity against *A. fumigatus*, with MICs ranging between 1 to 10 μg/mL [77,89] (Table 3).

### 3.3. A. fumigatus Effective Compounds from Marine Algae

Both compounds and extracts from several marine algae have been demonstrated to exhibit antifungal activities against *A. fumigatus.* The bromophenol compound 2,20,3,30-tetrabromo-4,40,5,50-tetrahydroxydiphenylmethane is one of the most potent compounds from marine algae, with a potent activity against *A. fumigatus* (with an MIC of 0.78 μg/mL) [94] (Table 4; Figure 3). Furthermore, ethanol extracts from *Laurencia majuscula* and *Padina pavonica* displayed an effective activity against *A. fumigatus* at low MICs of 1.95 and 0.95 μg/mL, respectively [71] (Table 4).

### 3.4. A. fumigatus Effective Compounds from Sea Cucumbers

Several triterpene glycosides (Figure 4) with good antifungal activities against *A. fumigatus* have been isolated and characterized from sea cucumbers such as *Holothuria scabra, Actinopyga lecanora,* several species of *Bohadschia*, etc. [80,81,96,97]. Among the best bioactive compounds isolated from these marine echinoderms, potent activities with an MIC_80_ ranging from 1.0 to 4.0 μg/mL have been recorded in in vitro screenings (Table 5). Triterpene glycosides therefore present very good prospects for future antifungal drug development.

Besides characterized compounds, both crude and partially purified extracts from different sea cucumbers have also been demonstrated to exhibit varying levels of antifungal activities against *A. fumigatus.* Ismail et al. [79] reported that crude and semipurified extracts from both aqueous body fluid extracts and methanolic wall extracts from *Holothuria polii* displayed varying activities against *A. fumigatus* in a concentration-dependent manner. Methanolic body-wall crude extracts showed better activities than those of aqueous body fluid. However, there was no difference in activities among the extracts after purification. Similarly, Adibpour et al. [98] also recorded an antifungal activity against *A. fumigatus* from *Holothuria leucospilota* body-wall and coelomic fluid extracts but none from its cuvierian organs’ extracts.

### 3.5. A. fumigatus Effective Compounds from Marine Fungi

A few studies have reported bioactive compounds from marine fungi against pathogenic molds. With reference to *A. fumigatus,* to the best of our knowledge, only the marine fungus *Phoma* sp. produced a characterized bioactive compound, YM-202204, with an IC_80_ of 12.5 μg/mL against *A. fumigatus* [69]. We propose that most fungi do indeed produce necessary antagonism metabolites against their fellow fungi if “they feel threatened” or if there is competition. That is why we have proposed several enhanced pathways (e.g., the coculture approach) for isolating and screening bioactive compounds (Figure 5 and Figure 6).

## 4. Challenges and Future Prospects

One of the biggest challenges facing the discovery of effective antifungal agents against *A. fumigatus* and other pathogenic microorganisms from a rich marine environment is the limitation on culturable organisms. It has been proven that the vast majority of microorganisms in nature cannot be isolated through the usual cultural techniques and are therefore labelled as unculturable microorganisms [100,101], where marine microbes predominate. Therefore, obtaining these viable but unculturable marine microbes by using several methods, including molecular approaches [102] and simpler techniques such as a long incubation with low nutrition [101], will certainly open a new vista of opportunities for discovering new antimicrobial agents. This may consequently lead to obtaining and evaluating novel metabolites from those unisolated and uncharacterized microorganisms, increasing the possibility of discovering potent antimicrobials, including antifungal agents against *A. fumigatus* (Figure 5).

Furthermore, the techniques and methods currently adopted in the isolation and screening of these bioactive compounds require upgrading in order to increase the chances of identifying new broad spectrum antimicrobial agents. For example, mainly focusing on extracellular components from these marine microorganisms seriously limits the global progress in this search. Considering the fact that marine organisms are unique, with special cellular features and components to survive in extreme and peculiar environments, it will be desirable to isolate and test intracellular and membrane-bound polysaccharides, peptides and lipids [103,104].

Moreover, conducting an initial screening of these compounds with predominantly in vitro assays might just be another limitation in this search. It has previously been demonstrated that certain compounds with weak in vitro antimicrobial activities could still be quite effective in *in vivo* applications [105,106,107,108]. Such compounds could have alternative mechanisms of “conquering the pathogen(s) of interest” beyond biocidal/biostatic activities. Other mechanisms than just the usual growth inhibition of pathogens, such as the modulation of the host immune system or blocking the production of virulence factor(s), are enviable attributes of new compounds that can only be discovered with *in vivo* assays [107,108,109,110]. Therefore, *in vivo* screening assays are highly recommended to discover bioactive compounds with broader modes of action.

Using invertebrate models such as *Caenorhabditis elegans* for a high-throughput *in vivo* screening has numerous advantages, including the absence of an ethical license requirement, reduced cost, labor and resources, as well as the possibility to simultaneously determine the cytotoxity of bioactive compounds [105,110]. We have recently established a *C. elegans*-based *A. fumigatus* infection model [111], making the high-throughput evaluation of the efficacy of bioactive compounds possible. Furthermore, we also discovered that the *in vivo* efficacy of some antifungal agents was different from what was observed in in vitro screenings, as demonstrated by the different killing modes of amphotericin B and the azole drugs. These therefore provide confidence in high-throughput *in vivo* assays as the primary screening approach.

Based on the general numbers of bioactive compounds from marine microbes, there are multiple strategies for inducing the production of some essential bioactive metabolites that possess antifungal properties, such as coculturing marine microbes with stimulators (other microorganisms, pathogens/their metabolites) or exposing them to stress conditions. Heavy metal stress could induce the expression of genes that lead to the production of desired metabolites, some of which have also been discovered to possess antimicrobial properties [112,113,114,115,116]. The use of heavy metals in the culture media of certain marine organisms is necessary for inducing the production of some essential metabolites, which may have antifungal properties. Evaluating such antifungal potency on *A. fumigatus* would really help to broaden the future search and discovery of antimicrobial compounds.

We have therefore suggested cultural approaches in Figure 6 in order to broaden and hasten the search for antifungal agents against *A. fumigatus*. Six culture approaches, ranging from monocultures to cocultures under different conditions, are suggested to stimulate genes involved in the production of uncommon metabolites, thereby increasing the possibilities of finding novel bioactive compounds that may otherwise not be identified through conventional approaches.

## 5. Conclusions

The search for antifungal agents effective against *A. fumigatus* from the marine habitat has not been all that productive (compared to other similar leading opportunistic pathogens like *Candida albicans*) for years now. A change of strategy is therefore indispensable and urgent if we must win the raging menacing war against this mold pathogen. Adoption of alternative cultural methods and evaluation of compounds for other treatment mechanisms would help to broaden the horizon of this search and may as well just be the needed breakthrough that we are waiting for.

## Figures and Tables

**Figure 1 antibiotics-09-00813-f001:**
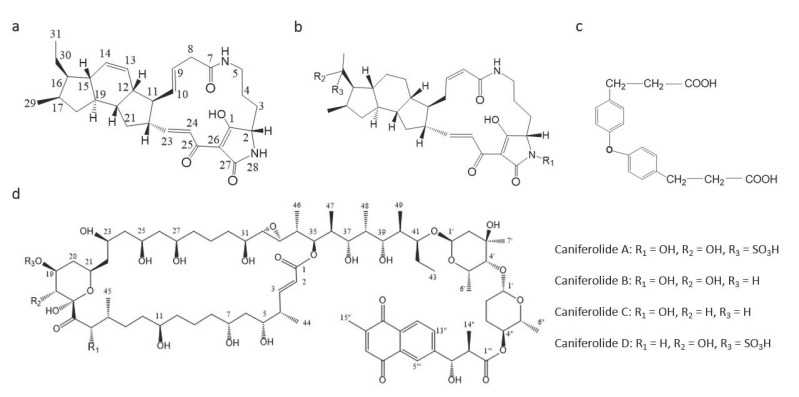
Chemical structures of compounds from marine bacteria with antifungal activity against *A. fumigatus.* (**a**) Isoikarugamycin, (**b**) 28-***N***-methylikaguramycin, (**c**) 4,4′-oxybis [3-phenylpropionic acid] and (**d**) Caniferolide A–D.

**Figure 2 antibiotics-09-00813-f002:**
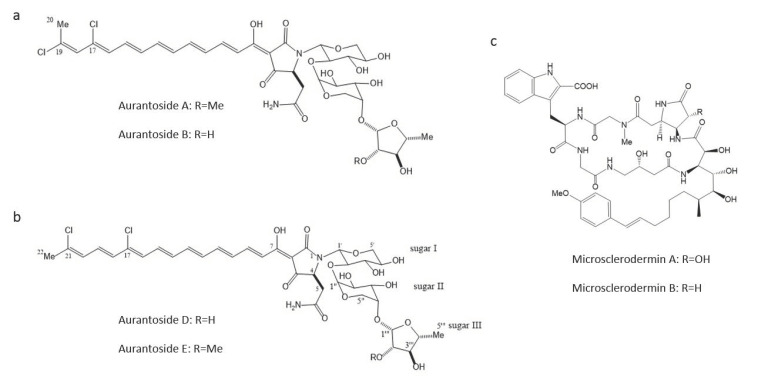
Chemical structures of compounds from marine sponges with antifungal activities against *A. fumigatus.* (**a**) Aurantosides A & B, (**b**) Aurantosides D & E, and (**c**) Microsclerodermin A & B.

**Figure 3 antibiotics-09-00813-f003:**
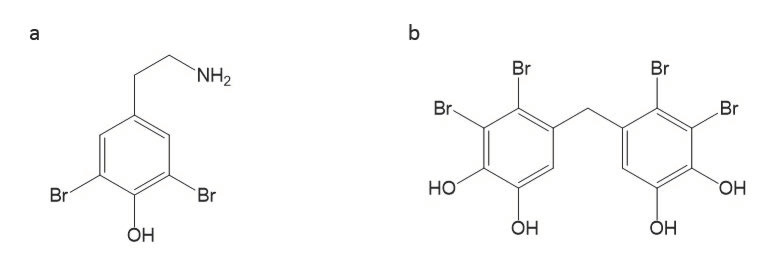
Bromophenol compounds from marine algae with antifungal activities against *A. fumigatus.* (**a**) 2,20,3,30-tetrabromo-4,40,5,50 tetrahydroxydiphenylmethane and (**b**) 3-bromo-4-(2,3-dibromo-4,5-dihydroxybenzyl)-5 methoxymethylpyrocatechol.

**Figure 4 antibiotics-09-00813-f004:**
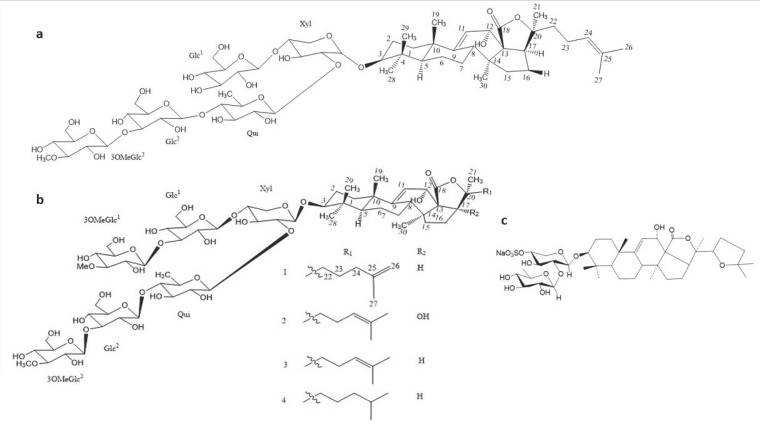
Chemical structure of compounds from sea cucumbers with effective antifungal activities against *A. fumigatus*. (**a**) Impatienside B, (**b**) Marmoratoside A (1); 17α-hydroxy impatienside (2); Impatienside A (3); & Bivittoside D (4), and (**c**) Holothurin B.

**Figure 5 antibiotics-09-00813-f005:**
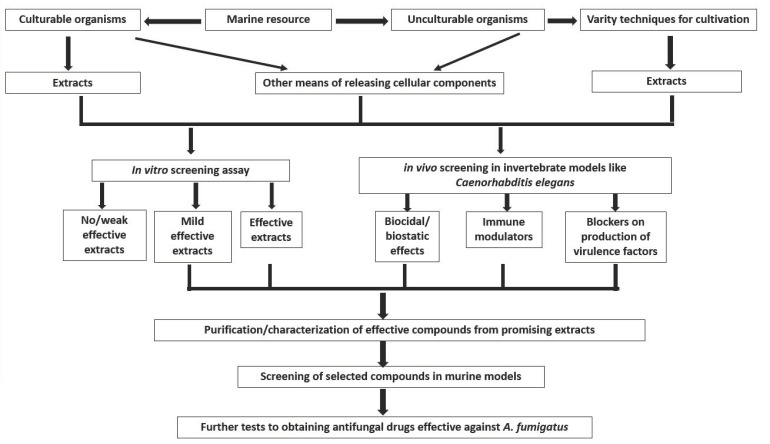
Flow chart of possible approaches to enhance the chances of discovering potent antifungal agents against *A. fumigatus.*

**Figure 6 antibiotics-09-00813-f006:**
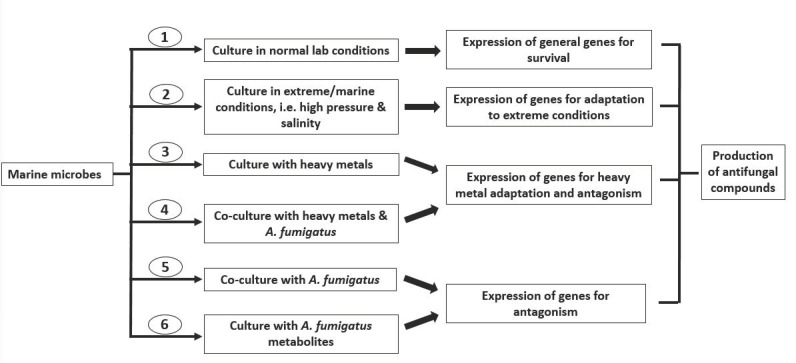
Suggested culture approaches to increase the discovery of antifungal bioactive compounds.

**Table 1 antibiotics-09-00813-t001:** Compounds from marine *Streptomyces* and other bacteria with antifungal activity against *A. fumigatus*.

Compound	Description/Source of Compound	Compound Class	Activity (MIC μg/mL)	Reference
Isoikarugamycin	Ethyl acetate extract from *Streptomyces zhaozhouensis* CA-185989 broth	polycyclic tetramic acid macrolactams	4–8	[83]
28-*N*-methylikaguramycin	Ethyl acetate extract from *Streptomyces zhaozhouensis* CA-185989 broth	polycyclic tetramic acid macrolactams	4–8	[83]
Ikarugamycin	Ethyl acetate extract from *Streptomyces zhaozhouensis* CA-185989 broth	polycyclic tetramic acid macrolactams	4–8	[83]
Caniferolides A&B	Ethyl acetate extract from *Streptomyces caniferus* CA-271066.	polyol macrolides	2–4	[85]
Caniferolides C&D	Ethyl acetate extract from *Streptomyces caniferus* CA-271066.	polyol macrolides	4–8	[85]
4,4′-oxybis(3-phenylpropionic acid)	Ethyl acetate concentrates of methanolic extract from *Bacillus licheniformis*	oxyneolignan	7–10 mm *	[65]

* Zone of inhibition given by 50 µg/disc.

**Table 2 antibiotics-09-00813-t002:** Antifungal activity of extracts from marine actinomycetes and other bacteria against *A. fumigatus* or MDR *A. fumigatus.*

Isolate	Description	Activity (MIC μg/mL)	Reference
*Streptomyces* sp. VITSDK36	Ethyl acetate extract	1.32	[87]
*Streptomyces* sp. VITSDK37	Ethyl acetate extract	0.74	[87]
*Streptomyces* sp. VITSDK38	Ethyl acetate extract	1.64	[87]
*Streptomyces* sp. VITSDK39	Ethyl acetate extract	1.78	[87]
*Streptomyces* sp. VITSDK41	Ethyl acetate extract	2.68	[87]
*Streptomyces* sp. VITSDK42	Ethyl acetate extract	1.58	[87]
*Actinopolyspora* sp. VITSDK43	Ethyl acetate extract	1.58	[87]
*Actinopolyspora* sp. VITSDK44	Ethyl acetate extract	2.54	[87]
*Micromonospora* sp. VITSDK46	Ethyl acetate extract	1.40	[87]
*Sachharopolyspora* sp. VITSDK47	Ethyl acetate extract	0.90	[87]
*Streptomyces* VITSVK5 spp.	Ethyl acetate extract	0.5	[86]
*Micrococcus* sp. RRL-3	Methanol extract	0.5	[64]
*Flavobacterium* sp. RRL-10	Methanol extract	5.5	[64]
*Pseudomonas* sp. RRL-12	Methanol extract	0.9	[64]
*Streptomyces* sp. RRL-13	Methanol extract	0.7	[64]
*Flavobacterium* sp. RRL-20	Methanol extract	4.3	[64]
*Flavobacterium* sp. RRL-32	Methanol extract	0.9	[64]
*Micrococcus* sp. RRL-37	Methanol extract	0.8	[64]
*Bacillus* sp. RRL-38	Methanol extract	2.0	[64]
*Flavobacterium* sp. RRL-41	Methanol extract	1.3	[64]
*Flavobacterium* sp. RRL-54	Methanol extract	0.5	[64]
*Alcaligenes* sp. RRL-56	Methanol extract	0.3	[64]
*Bacillus* sp. RRL-57	Methanol extract	2.5	[64]
*Streptomyces* VITSVK5 spp.	Ethyl acetate extract against MDR9	4 *	[86]
*Streptomyces* VITSVK5 spp.	Ethyl acetate extract against MDR10	0.25 *	[86]
*Streptomyces* VITSVK5 spp.	Ethyl acetate extract against MDR11	2 *	[86]

* Against multidrug resistant *A. fumigatus* strain.

**Table 3 antibiotics-09-00813-t003:** Effective compounds against *A. fumigatus* from marine sponges.

Compound	Description/Source of Compound	Location of Isolation	Compound Class	Activity (MIC μg/mL)	Reference
Swinhoeiamide A	Methanol extract from *Theonella swinhoei*	Karkar Island, Papua New Guinea	calyculinamide-related congener	1.0	[89]
1,2-dioxane ring peroxide acid	Ethanol extract from *Plakortis halichondrioides*	Long Bay, Negril, Jamaica	peroxide	5.6 ^1^	[90]
Aurantoside A & B	Ethanol extract from *Siliquariaspongia japonica*	Hachijo-jima Island, Tokyo, Japan	polyene tetramic acids	0.16	[88]
Aurantoside D	Ethanol extract from *Siliquariaspongia japonica*	Hachijo-jima Island, Tokyo, Japan	polyene tetramic acids	11.0 ^2^	[88]
Aurantoside E	Ethanol extract from *Siliquariaspongia japonica*	Hachijo-jima Island, Tokyo, Japan	polyene tetramic acids	0.04	[88]
3,5-dibromo-2-(3,5-dibromo-2-methoxyphenoxy) phenol	Ethyl acetate-methanol (EtOAc-MeOH) extract of *Dysidea herbace*	Coast of Zanzibar	phenol	7.8	[91]
Microsclerodermin J	Methananol extract of *Microscleroderma herdmani*	Mauritius	hexapeptides	10.0	[77]
Microsclerodermin K	Methananol extract of *Microscleroderma herdmani*	Mauritius	hexapeptides	10.0	[77]
Microsclerodermin A	Methananol extract of *Microscleroderma herdmani*	Mauritius	hexapeptides	1.3	[77]
Microsclerodermin B	Methananol extract of *Microscleroderma herdmani*	Mauritius	hexapeptides	0.6	[77]
Araguspongin C	Methanol extract of *Haliclona exigua*	Tamil Nadu coast of India	heteropentacyclic *compound*	50	[75]
Spongistatin 1	Extract from *Hyrtios erecta*	Eastern Indian Ocean	macrocyclic lactone polyether	12.5 ^3^	[92,93]

^1^ IC_90_; ^2^ inhibition zone (mm); ^3^ minimum fungicidal concentration (MFC).

**Table 4 antibiotics-09-00813-t004:** Effective antifungal compounds/extracts against *A. fumigatus* from marine algae.

Compound/Isolate	Description/Source of Compound	Activity (MIC μg/mL)	Reference
2,20,3,30-tetrabromo-4,40,5,50-tetrahydroxydiphenylmethane	Compounds from *Odonthalia corymbifera*	0.78	[94]
3-bromo-4-(2,3-dibromo-4,5-dihydroxybenzyl)-5-methoxymethylpyrocatechol	Compounds from *Odonthalia corymbifera*	25	[94]
(*E*)-2-{(*E*) tridec-2-en-2-yl} heptadec-2-enal	Chloroform/methanol extract of *Laurencia papillosa*	200	[95]
Extract	Ethanol extract from *Laurencia catarinensis*	3.90	[71]
Extract	Ethanol extract from *Laurencia majuscula*	1.95	[71]
Extract	Ethanol extract from *Padina pavonica*	0.98	[71]

**Table 5 antibiotics-09-00813-t005:** Characterized compounds against *A. fumigatus* from sea cucumbers.

Compounds	Sea Cucumber	Compound Class	Activity (MIC μg/mL)	Reference
Scabraside A	*Holothuria scabra*	Triterpene glycosides	2.0 *	[97]
Echinoside A	*Holothuria scabra*	Triterpene glycosides	1.0 *	[97]
Holothurin A_1_	*Holothuria scabra*	Triterpene glycosides	8.0 *	[97]
Holothurin B	*Actinopyga lecanora*	Triterpene glycosides	3.12	[82]
3-O-b-D-xylopyranosyl-16b-acetoxyholost-7-ene	*Actinopyga lecanora*	Triterpene glycosides	50.0	[96]
Holothurin A	*Actinopyga lecanora*	Triterpene glycosides	50.0	[96]
Bivittoside-D	*Bohadschia vitiensis* Semper	Lanostane triterpenoid	1.56	[99]
Marmoratoside A	*Bohadschia marmorata* Jaeger.	Triterpene glycosides	2.81 *	[80]
Marmoratoside B	*Bohadschia marmorata* Jaeger.	Triterpene glycosides	44.44 *	[80]
17α-hydroxy impatienside A	*Bohadschia marmorata* Jaeger.	Triterpene glycosides	2.78 *	[80]
Impatienside A	*Bohadschia marmorata* Jaeger.	Triterpene glycosides	2.81 *	[80]
Bivittoside D	*Bohadschia marmorata* Jaeger.	Triterpene glycosides	2.80 *	[80]
25-acetoxy bivittoside D	*Bohadschia marmorata* Jaeger.	Triterpene glycosides	43.13 *	[80]
Arguside F	*Holothuria (Microthele) axiloga*	Holostan-type triterpene glycosides	16.0 *	[81]
Impatienside B	*Holothuria (Microthele) axiloga*	Holostan-type triterpene glycosides	4.0 *	[81]
Pervicoside D	*Holothuria (Microthele) axiloga*	Holostan-type triterpene glycosides	64.0 *	[81]

* MIC_80_.

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
