# Peer review of "Marine Bioactive Compounds against *Aspergillus fumigatus*: Challenges and Future Prospects"

_antibiotics, 2020, doi:10.3390/antibiotics9110813_

Round 1

Reviewer 1 Report

A nice review article covering an area that needs information of this type. I made some minor suggestions on the attached PDF, most of which were minor English sentence construction.

However, I would suggest removing Table 3 as any compound that has an MIC of >64 is effectively inert in such an assay.

Author Response

  1. A nice review article covering an area that needs information of this type. I made some minor suggestions on the attached PDF, most of which were minor English sentence construction. Answer: Thanks to the reviewer. The indicated corrections have been effected in the revised manuscript.
  2. However, I would suggest removing Table 3 as any compound that has an MIC of >64 is effectively inert in such an assay. Answer: Table 3 has been removed in the revised manuscript.
  3. It is probable that this is from a microbe in the sponge. Need to consider this as shown by the work of Piel et al., 2014. Answer: We recognized that marine organisms such as sponges have several microbial flora including those in symbiotic relationship (as illustrated by Piel et al., 2014; Lackner et al., 2017 amongst other works) and those in loose association. We have considered in our review as a number of bacterial isolates (including actinomycetes) were in association and isolated from several marine organisms including marine sponges [50,51,53], which were captioned under “3.1. A. fumigatus effective compounds from marine bacteria.” The Swinhoeiamide A presented by Edrada et al. [77] was isolated directly from Papua New Guinean sponge Theonella swinhoei and not from a microbe in the sponge form.
  4. I would suggest rechecking the zone size as this is over 4 metres. Answer: We have rechecked the data in the original paper and didn’t find clear statement for this parameter. Therefore we have removed this reference in the revised manuscript.

Reviewer 2 Report

The review article by Ahamefule et al. summarises studies that have reported bioactive compounds that target the potential human pathogen and opportunistic fungus Aspergillus fumigatus. In this manuscript the authors focus upon anti-fungal compounds isolated specifically from the marine environment. They propose future directions to help other researchers investigating culturing of marine-derived organisms and compound extraction.

This is a generally well written manuscript, which is easy to read, that curates numerous studies on this topic into one resource that will be useful to those working in this defined field. I have attached a draft containing minor corrections to improve flow and clarity. There are a few areas that need some attention.

In Tables 1, 4 and 6 - change "Types of compounds" to "Chemical Class"

Table 2, move the * next to the MIC value

Table 3 - all compounds have MIC greater than 64 ug/ml. Were these indeed active? A value of >64 might suggest that they have no anti fungal activity. Can this be clarified further?

Table 4, sub note 3. What is MFC?

Author Response

  1. In Tables 1, 4 and 6 - change "Types of compounds" to "Chemical Class"

Answer: The indicated changes have been made in Tables 1, 4 and 6 in the revised manuscript.

  1. Table 2, move the * next to the MIC value

Answer: The * has been moved to the MIC values in Table 2 in the revised manuscript.

  1. Table 3 - all compounds have MIC greater than 64 ug/ml. Were these indeed active? A value of >64 might suggest that they have no antifungal activity. Can this be clarified further?

Answer: Table 3 has been removed in the revised manuscript.

  1. Table 4, sub note 3. What is MFC?

Answer: The sub note means minimum fungicidal concentrations. It has been included in the table in the revised manuscript.

Reviewer 3 Report

This review mostly tabulates MICs of marine-derived natural products against Aspergillus fumigatus. On the negative side, the insights offered are minimal, the mechanisms of action are generally unknown, and most of the compounds are highly complex and not easily modified in the search for more effective potential drugs. On the positive side, the review is potentially valuable as a reference. As such, information on mass yields would be useful for these potential lead compounds, wherever it is available.

Specific suggestions by line number

41 Epidemiological evidence that drug-resistant infections are increasing should be cited, or the statement should be dropped.

58-61 It would be good to summarize some of the specific data on infections in non-immunocompromised patients.

63 “evolvement” should be “evolution” or ”increase”

Table 3: It is not clear that this material adds to the review. What is the evidence that these compounds are ‘bioactive’ when the MICs are above the maximal tested concentrations.

197-198: In vivo screening for initial activity seems to me to be too expensive and time-consuming to be practical. Perhaps the authors have specific inexpensive models in mind, and if so they should be mentioned here.

Author Response

  1. This review mostly tabulates MICs of marine-derived natural products against Aspergillus fumigatus. On the negative side, the insights offered are minimal, the mechanisms of action are generally unknown, and most of the compounds are highly complex and not easily modified in the search for more effective potential drugs. On the positive side, the review is potentially valuable as a reference. As such, information on mass yields would be useful for these potential lead compounds, wherever it is available.

Answer: The mass yield of some of the lead compounds were not given in the literature. For some others that were given, we discovered that different fractions could give the same active compound (with different mass) at different time and purification level. It was therefore challenging to compile the mass yields of those compounds.

  1. 41 Epidemiological evidence that drug-resistant infections are increasing should be cited, or the statement should be dropped.

Answer: We have given epidemiological evidences that drug resistant A. fumigatus infection (aspergillosis) is increasing in the revised manuscript.

  1. 58-61 It would be good to summarize some of the specific data on infections in non-immunocompromised patients.

Answer: We have included several examples of reported varying kinds of aspergilloses in the revised manuscript.

  1. 63 “evolvement” should be “evolution” or ”increase”

Answer: “Evolvement” has been corrected to “evolution” in the revised manuscript

  1. Table 3: It is not clear that this material adds to the review. What is the evidence that these compounds are ‘bioactive’ when the MICs are above the maximal tested concentrations.

Answer: Table 3 has been removed from the revised manuscript.

  1. 197-198: In vivo screening for initial activity seems to me to be too expensive and time-consuming to be practical. Perhaps the authors have specific inexpensive models in mind, and if so they should be mentioned here.

Answer: Indeed the conventional in vivo model is quite expensive, complex and also cumbersome to be applied in initial in vivo screenings. However, we have recently demonstrated the ease and inexpensive nature of adopting Caenorhabditis elegans as the in vivo model for evaluating effective agents against nematode aspergillosis [111]. It is this in vivo model that we recommend in the manuscript.